# Sustainability of Online Teaching: The Case Study Mother Tongue Spelling Course at Montenegrin Universities

**Milena Buric** [1,*] , **Milijana Novovic Buric** [2] , **Andjela Jaksic Stojanovic** [3] , **Ljiljana Kascelan** [2] and **Dragica Zugić** [4]

1  Faculty of Philology, University of Montenegro, 83000 Niksic, Montenegro
2  Faculty of Economics Podgorica, University of Montenegro, 81000 Podgorica, Montenegro
3  Faculty of Culture and Tourism, University of Donja Gorica, 81000 Podgorica, Montenegro
4  Faculty of Philology, University of Donja Gorica, 81000 Podgorica, Montenegro
*  Correspondence: milenab@ucg.ac.me

**Abstract:** This paper aims to examine the sustainability of online teaching using the ZOOM application. The research is exclusively related to the course entitled Mother Tongue Spelling at the university level. Its main goal is to analyze the perceptions of surveyed students of Montenegrin philology faculties about the advantages and disadvantages of this type of teaching and their attitude towards traditional, online or blended models of teaching. This paper will examine the opinion of students regarding the application of dialogic, monologic, textual and blended communication methods. Descriptive statistics and the decision tree model were used in processing the survey data. The research showed that students see the availability of lecture recordings and attending lectures regardless of their current location as the biggest advantages of online spelling classes, while the most significant limitations point to technical problems and lack of social interaction. The results also showed that the students prefer the application of the blended method, while the monological method was rated the lowest. The key contribution of this paper is its focus on online learning of the students' mother tongue, not a foreign language.

**Keywords:** online teaching and learning; Zoom application; sustainable education; blended approach; mother tongue; literacy acquisition

## 1. Introduction

Over the last two years, the education system at the global level has been exposed to sudden and very turbulent changes caused by the planet-known virus. Almost overnight, the shift from the traditional to the online format of work at all levels of education took place [1]. Most universities did not even have all the necessary infrastructure or support services enabling them to effectively conduct the online teaching and learning process [2]. Additionally, some universities, especially in developing and non-developed countries, were faced with a lack of digital knowledge, skills and competence in teaching staff, who were unprepared for this kind of process; therefore, their transition from face-to-face to online teaching happened without any preparation, training or support [3]. All this initially harmed the quality of online classes, and thus, the very process of students' acquisition of knowledge from many areas, including the spelling of their mother tongue. Furthermore, it was noticed that to start with digital teaching, a basic level of digital competence is required, and to implement quality online teaching, digital competence must be at an advanced level [4]. In an emergency, i.e., a crisis caused by an epidemic that implied physical isolation, it was not easy to successfully organize the course on how to teach online using the tools offered by digital platforms, nor were many teachers interested in improving the aforementioned skills [5]. Additionally, national and institutional legislation in many countries, as well as in Montenegro, did not recognize online learning, and those who recognized it did not have adequate procedures to implement it. Special problems

occurred regarding the organization of online exams, which also were not recognized by the legislation of most countries, especially in the Western Balkan region. On the other hand, online tools provide the opportunity for students' work to be accessed by an unlimited audience, which increases students' motivation but also responsibility for the content of the work itself [6].

Similar to other developing countries, at the beginning of COVID-19 pandemic, Montenegrin higher education institutions had to face many challenges in the implementation of online teaching [7,8]. The educational process had to be transited from face-to-face to online in a very short period of time, and there was a lack of financial resources that would enable the use of certain online platforms that have already been widely used by developed countries in the implementation of the teaching process. Additionally, most teaching staff and students were not very familiar with the E-learning methodology such as: online learning portals, video conferencing and streaming on social media, such as WhatsApp or Telegram [7,9]. Furthermore, apart from IT skills and competence, similar problems appeared as in other developing countries, such as lots of students who live in remote areas without internet connections [10], those coming from poor families attending online classes with many family members in the same room and lack of educational tools, such as laptops, mobile phones, tablets, etc. [8].

Because of all the above, educational institutions in Montenegro decided to use the ZOOM application with which they had the most experience in the period before the pandemic, and the application was free. One of the advantages of the Zoom application for educational purposes is the possibility for simple and quick involvement of students not only via computer but also via phone, which implies a new possibility, i.e., monitoring classes from any location with internet coverage. Furthermore, an important addition is a tool for recording the entire lecture. The benefit is mutual, because students can watch the same lecture more than once if they need it for more successful mastering of the learning material, and the professor can see his/her possible methodological oversights and other mistakes by reviewing the video sometime later in order to avoid them and, thus, improve the teaching process itself. The Zoom application also enables teaching through the application and a combination of traditional communication methods, monologue, dialogue and text, with the integration of materials available online. On the other hand, one of the biggest shortcomings is the questionable credibility of student answers to questions and assignments. The professor has almost no control over the individual work of the student, who can turn off the camera and in response use materials that he/she did not prepare himself/herself. In addition, the use of the Zoom application for educational purposes is not legally regulated in most countries, so the professor is not familiar with the procedure and measures to sanction the non-academic misconduct of students during this type of teaching. Additionally, an important disadvantage of using the ZOOM application is its possible abuse, such as students' joining from inappropriate places (city transport, restaurants, public places, etc.).

This paper intends to investigate the influence of online learning on the process of acquiring knowledge of mother tongue spelling, which is in the native language of the students surveyed, paying attention to the most important thematic topics in this area, as well as to different methodological approaches. The mother tongue is the first language a person learns in childhood, the language of the family and the country in which he lives, and as such, is part of his personal, social and cultural identity. Knowledge of the mother tongue does not only have a positive effect on learning foreign languages but is also a prerequisite for the development of cognitive abilities and intellectual growth of a person [11].

In Montenegro, according to the Constitution from 2007. the official languages are Montenegrin, Serbian, Croatian and Bosnian [12], which were created after the dissolution of the Serbo-Croatian language community, and which their codifiers tried to shape as special, new linguistic identities, even though it is the same language system [13]. Since Montenegrin school programs contain a cumbersome, four-name name for the mother

tongue: Montenegrin—Serbian, Croatian, Bosnian, or only Montenegrin [14], in the paper, we prefer the term *mother tongue* as a general term, i.e., as a hyperonym.

Spelling is most often defined as a generally accepted and unique system of rules for writing words specific to one language, but it also implies the art of writing according to accepted norms [15]. These norms include prescribing and prohibiting [16], and their knowledge and application are the main conditions of literacy. Regardless of the name (Serbo-Croatian, Serbian, Montenegrin, Bosnian or Croatian), the spelling rules in the mentioned languages are based on the rule that one letter corresponds to one voice, which is not the case in other languages, and the rules that regulate writing apply to speech, which further simplifies learning. Because of that fact, spelling is studied in Montenegro in primary and secondary schools and only in certain study programs of state and private universities. We believe that things should be changed in this respect, and that the mother-tongue spelling course should be incorporated in all study programs of all universities, because literacy, that is, language competence, is one of the most important components of professional development [6,17].

According to the adopted syllabi of *Mother Tongue Spelling,* the most important teaching units are: writing capital letters, joint and separate words, abbreviations, voice alternations, writing consonants *j* and *h*, loss of consonants, spelling and punctuation marks, etc., as indicated by the reference literature [18–22]. There are four communication methods: dialogical, monologue, text and blended. These methods represent a way of transmitting information in educational work and are based on the language as a system of signs for communication [23]. The dialogical method enables the acquisition of knowledge and skills through conversation. The dialogic form of communication is not just a matter of asking questions and answering. Psychologically, the dialogic form of communicating also differs in that each partner's response depends on the other's speech behavior. This entails the need for a quick response, which should also be taught. At the heart of the monologue method is a complete and systematic oral presentation of one actor in the classroom, either a professor or a student, which involves thorough preparation and a timely division of responsibilities. The text method is based on the use of written or printed words in the process of learning and is abundantly represented in the teaching of the mother tongue in the form of reading the achievements of linguistic sciences (consulting dictionaries, encyclopedias, lexicons) [23]. The blended method is the result of the application of methodological pluralism, which breaks the uniformity of lectures, which is extremely important, especially having in mind the fact that students' curiosity is not only stimulated by the course itself, but even more by the way of teaching. Experience shows that in the traditional teaching of the mother tongue (and thus spelling), methods are supplemented by a skillful combination with a blended method, and their parallel, successive or alternating application strengthens the effectiveness of teaching [23]. Zoom has gained a very wide affirmation in the field of education precisely because it offers the tools that support these methods immanent to traditional teaching, as well as their combination [24]. Online teaching of Mother Tongue Spelling does not have its own methodology but relies on the methodology that is established in traditional teaching and is based on the application of these communication methods.

This is also the first study that analyzed the impact of online teaching on acquiring knowledge of the mother tongue spelling, not only in Montenegro but also in the Western Balkans region. Furthermore, this work aims to establish the level of students' satisfaction with ZOOM classes as well as their perception of the communication methods applied during this type of class, which examines the possibility of sustainability of online teaching the Mother tongue spelling in the post-pandemic period. The realization of the set goals is analyzed based on the information obtained through an online survey.

Considering the fact that the Mother tongue spelling is a one-semester course taught mainly at faculties of philology in Montenegro, with outcomes such as raising the level of literacy, i.e., enhancing the culture of speech of the population through higher education, it is limited to only a small number of study programs. In general, the culture of speech

represents the speaker's ability to effectively use linguistic means of expression in a given situation, and its cultivation and improvement depend on knowing the orthographic rules of the native language [25].

Therefore, our paper can motivate the management and professional bodies of Montenegrin universities to introduce a one-semester course entitled Mother Tongue Spelling during the accreditation of new and re-accreditation of existing study programs, which would be conducted fully or partially online in all study programs. In this way, the sustainability of online teaching mother tongue spelling would be ensured and it would also permanently increase the level of the speaking culture of the population. Our research can be an incentive for domestic authors, as well as authors from the Western Balkan region and beyond, to research similar topics. This would contribute to supplementing the literature in the area of the effects of applying communication methods in the online teaching of other social disciplines.

Finally, we believe that our paper can contribute to the destigmatization of online teaching as of lower quality compared to traditional teaching by pointing out its advantages but also the effects of the application of communication methods that are common to them.

## 2. Literature Review

When it comes to sustainability, education is considered a tool that can be used to address sustainability challenges. "The key to creating a more sustainable and peaceful world is learning" [26]. This perspective is most often termed as education for sustainable development (ESD), sustainability education (SE), and education for sustainability (EfS), with ESD as the most popularly used. A UNESCO definition of education for sustainable development is "an interdisciplinary learning methodology covering the integrated social, economic, and environmental aspects of the formal and informal curriculum" [27].

Another interpretation of sustainability [28] that relates to the aim to disseminate effective e-learning practices beyond the development context considers an e-learning initiative sustainable when all three of these conditions are met:

- A learning design involving information and communications technology was developed and implemented within a course or courses of study.
- The e-learning concept, design, system or resources have proven potential to be adopted, and possibly adapted, for use beyond the original development environment.
- Maintenance, use and further development of the e-learning concept, design, system or resources do not remain dependent on one or a few individuals who created them, to the extent that, if their involvement ceased, future prospects would not be compromised.

The recent literature describes a number of general principles of sustainability pedagogies, including participatory and inclusive education processes, transdisciplinary, cooperation and experiential learning; all of which involve student-centered and interactive approaches to teaching and learning [10].

Previous studies of online teaching have mainly focused on learning a foreign language [6,17,29] and not on the mother tongue, thus, our research fills a gap in the contemporary literature.

One of the studies that deals with technology in the context of learning English emphasizes the need to train teachers to use e-resources, as well as the necessity of integrating technology and traditional teaching [30]. According to this research, by using online materials (via computers or smartphones), students are relieved of the burden of using their textbooks, and learning becomes an interesting activity experienced not as an obligation but as a game, thus, maintaining their interest in learning English through an informal educational process.

Similar conclusions were reached by the author of the research on the online teaching of the mother tongue, who claims that for the successful application of technology, the digital, professional and methodical competences of teachers are necessary, as well as adequate legal regulations that will define the ways of implementing online teaching in

the traditional educational process. She also emphasizes that such experience shows that during the online teaching of the mother tongue through the Zoom application, the two-way nature of the teaching process was intensified: teachers and students learned from each other, i.e., teachers as members of a more mature population had the opportunity to learn from students whose digital competence is at a more advanced level, since the youth population has been immersed in a technological society since birth [24]. This resulted in establishing a special kind of closeness, a rapport in communication between professors and students [31], which significantly mitigated the negative psychological consequences caused by the lack of social interaction during the pandemic.

Having analyzed the recent studies that examined students' attitudes about online learning in general [32–34], it is easy to notice the lack of such studies in Western Balkan countries, especially regarding higher education. Furthermore, only a few studies focused on the examination of the students' results and achievements in different areas, especially in art and humanities. Students who learn online face several challenges due to the struggle to completely adapt to online courses and the lack of interaction between students and their tutors [32]. E-learning platforms motivate student-centered learning, and they are easily adjustable during abrupt crises, such as COVID-19 [32]. Students are comfortable with online classes and are obtaining enough support from teachers, but they do not believe that online classes will replace traditional classroom teaching [16]. In addition to these, a recent study conducted in Bangladesh [34] discovered that female students seemed to be more positive about online teaching than male students, whereas urban students had more positive appreciation than rural students.

Previous research on online teaching and the learning process was mostly related to the teaching and learning process in general [4]. In some developing and non-developed countries, students have faced some infrastructural problems, such as lack of space for learning, lack of access to the Internet, high costs of using the Internet, etc. [35,36].

In Montenegro, as in other countries of the Western Balkans, research shows that in primary and secondary schools, the most dominant tools used for carrying out the teaching and learning process are Google classrooms and online models such as Viber groups. Many world higher education institutions, including Montenegrin, have chosen the Zoom application as an alternative to face-to-face teaching, giving it an advantage over other software [37,38].

Different studies related to the level of education of students were conducted at the national and international levels [39–42]. Most of them focused on the influence of different sociodemographic factors, such as gender, level of studies and student status, on students' general attitudes toward online learning and identification of the main advantages and disadvantages of this process [43–45], which we also deal with in this paper, but exclusively from the aspect of acquiring the knowledge of Mother Tongue Spelling by Montenegrin university students.

According to the results of the previous research, it may be concluded that the most common limits of the Zoom application are the following: lack of direct communication, questionable objectivity of assessment, lower educational efficiency compared to traditional teaching, technical problems, concentration span, and lack of digital competencies [4,39,46]. Among the most significant benefits are reduced study costs, improved digital skills, asynchronous communication, as well as the fact that the ability to record Zoom meetings has made teaching reusable [37,38,47]. Respondents also stated that the fact that the lecture does not depend on the location of the participants in the teaching and learning process is one of the most important benefits of the Zoom application [37,48].

One of the most relevant studies was conducted in June 2020, and it included 30,383 students from 62 countries [39], and it pointed out some of the main disadvantages of online learning, such as a lack of place for studying, lack of digital competence in teaching staff, lack of adequate IT infrastructure, difficulties in understanding materials, lack of time for preparation of materials, etc. [39–42,49,50]. Some studies pointed out the emotional distress of students and their negative emotions, such as anxiety, anger, shame,

frustration, etc., as consequences of the transition from face-to-face to online learning [39,42]. Numerous studies have concerned the lecturers' perception of the effect of online classes. Thus, the research that was focused on the analysis of the perception of teachers of higher education institutions about the use of virtual platforms for learning during the so-called extraordinary (or emergency) remote teaching (ERT = emergency remote teaching) showed that teachers consider this type of teaching to be average or less effective, and their attitude depended on the level of training in using tools offered by virtual platforms and on age [51]. It was stated that with the increase in the age of teachers, their perception of the ability to manage virtual platforms decreased, as well as that perception did not depend on the chosen platform [51]. The group of authors who studied the competence necessary for a sustainable digital society pointed out: "Literacy in general, and digital literacy in particular, is a powerful tool to empower individuals and to equip them with competencies that will allow them to have a successful personal and professional life" [52]. Therefore, they not only emphasize the importance of digital but also general literacy for a sustainable society, which is emphasized in this paper.

When it comes to the philology field of studies, research on the success of online teaching was mainly focused on the effects of learning a foreign language, especially English [47], while there is an obvious lack of research regarding the teaching and learning of the students' mother tongue and its sustainability. According to the authors of a study on students' perception of online learning [32], online teaching does not have the power to attract students' attention. They also point out that, if the online class is long, students start to become bored and distracted. Since the student's lack of focus and interest in the class is not directly caused by the form of teaching (online or traditional) but by the inadequate application of communication methods, most often monologic [32], the goal of this paper is to examine the students' perception of the type of communication method used by the teacher of *Mother Tongue Spelling.*

More precisely, this paper examines the problem of implementing the Zoom application in online teaching and learning the mother tongue in all higher education institutions in Montenegro, with a special focus on spelling acquisition as the single-most important part of education.

## 3. Research Questions, Materials and Methods

### 3.1. Research Methodology and Research Questions

The aforementioned literature review regarding online teaching and communication methods indicates a lack of research in the case of *Mother Tongue Spelling*. Furthermore, the attitude of students, apart from the socio-demographic characteristics recommended by previous research, may also depend on the type of university they are studying at (state or private), degree of study (undergraduate or master), student status (full or part time), online teaching experience and class attendance, which we will define in this study as student profiles. Investigating the connection between the students' profiles and their opinion provides a deeper insight into the validity and causes of their attitudes, which has not been investigated so far and can be useful for improving the teaching process in general. In that sense, the aim of this paper is to obtain answers to the following four research questions:

RQ1: Is there and, if yes, what is the relationship between the students' profiles and their perception of the advantages of online teaching Mother Tongue Spelling with Zoom?

RQ2: Is there and, if yes, what is the relationship between the students' profiles and their perception of the shortcomings of online teaching Mother Tongue Spelling with Zoom?

RQ3: Is there and, if yes, what is the relationship between the students' profiles and the methods they prefer during online classes with Zoom related to Mother Tongue Spelling (dialogic, monologic, textual, blended)?

RQ4: Is there and, if yes, what is the relationship between the students' profiles and the model of teaching Mother Tongue Spelling (traditional, via Zoom or blended)?

The decision tree (DT) method was used to answer research questions RQ1–RQ4. This method was chosen because survey questions are based on multiple choice without the use of a scale, which generates discrete categorical variables (with a limited number of values—categories) for statistical analysis. Considering that the dependent variables are also of a discrete type, classification methods, such as classification trees, are imposed as effective for the analysis of this type of data. These methods, unlike regression methods, do not require preconditions regarding the distribution of input and output variables and their relationship (normality, correlation, multicollinearity, heteroscedasticity), as well as assumptions about the relationships between variables, i.e., specifying the functional form [53,54]. Classification methods are increasingly used for the analysis of survey data, especially the DT method [53–57].

Another reason for using this method is the very nature of the research questions, namely, it is necessary to analyze the relationship between independent variables that determine the student's profile (gender, state/private university, student status—full or part time, study level, online class experience, attending online classes) and dependent variables (advantages of online classes, disadvantages of online classes, preferred communication method and preferred instructional mode). The DT method identifies connections between dependent and independent variables through the so-called structure of the tree, i.e., nodes, branches and leaves. Each node in the tree is associated with one of the independent variables and each branch of the tree with a subset of the values of the corresponding independent variable. If the target variable is discrete (has a limited number of values), then the tree is considered a classification tree, and each leaf represents one value of the target variable (one class) [58]. The paths from the root to the leaves of this tree define if–then rules that describe classes in a simple and easy-to-understand way. As expected, all rules do not have the same importance, which is conditioned by the number of examples of the class represented by that leaf. The DT algorithm clearly determines relative frequencies for each leaf, i.e., the percentage of examples of the class represented by the leaf in relation to the total number of examples in the leaf (rule accuracy). This study will take into account the most important rules that describe the considered key outputs based on the factors that define the student's profile.

By the tree induction process, the initial data set is divided into subsets of sub-data with as little entropy as possible, i.e., to contain as many examples of one class as possible (defined by the values of the target variable). The degree of purity of the subsets obtained in this way is measured by one of the goodness of splitting measures, such as information gain, gain ratio or gini index [59,60]. Given that the information gain measure is biased towards predictors with a larger number of values [61], the gain ratio measure was used in this paper. The tree induction process is shown graphically in Figure 1 [58].

The initial data set is divided by the values of each individual attribute (predictor) and the quality of the division is calculated for each of the attributes. The attribute that gives the best split becomes the node of the tree. On each subset of the data, the procedure is repeated recursively, thus, creating nodes at a lower level of the tree. Tree splitting and growth are stopped when one of the stop conditions is met (all instances in the leaves belong to one class, the maximum tree depth is reached, or some other user-defined condition) [62]. Therefore, to reduce complexity, we set the maximum depth of the tree.

The number of correctly classified examples in relation to the total number of examples represents the overall accuracy of DT classification, while the number of misclassified examples in relation to the total number of examples represents classification error. Two important elements that define DT classification accuracy are: class precision (the number of accurately classified instances of one class in relation to all the instances that the model classified in that class), and class recall (the number of members of the class that the model correctly classified in relation to all members of that class) [56].

## Decision tree algorithm

**Figure 1.** The used tree induction algorithm.

The survey regarding the use of Zoom in teaching and learning the Mother Tongue Spelling was conducted online and anonymously. The questionnaire was sent via free communication platforms, i.e., smartphone applications—Viber—and email to students of the Faculty of Philology of the University of Montenegro, as well as to the students of philological faculties from private universities in Montenegro—University of Donja Gorica. In order to ensure the validity of the sample, the questionnaire was sent to students of all faculties in Montenegro which offer the *Mother Tongue* course as a part of their curriculum in both undergraduate and master studies. A total of 145 students responded to the survey (there were no invalid answers since the survey had built-in protections against incorrect input). The sample is representative with a confidence level of 95% and a margin of error of 7%, having in mind the fact that, according to official data in Montenegro, there are around 600 students in Montenegro that attend the course entitled Mother Tongue Spelling at undergraduate and/or post-graduate studies.

Based on the presented previous research and many years of experience in teaching *Mother Tongue*, multiple choice survey questions were designed and structured into four

related unit parts (see Appendix A). Choosing only one answer option reduces the number of input variables and the complexity of the DT model. Research questions that seek to determine the most important advantages, disadvantages, methods and way of teaching *Mother Tongue* with the ZOOM platform justify multiple choice survey questions and the choice of one answer option. Considering that the questions are of the multiple choice type, no scale was used as a measuring instrument.

The first part examines the socio-demographic characteristics of the students, i.e., gender, type of university (state/private), student status (full/part-time), level of study (undergraduate/master), as well as their participation in Zoom classes during the period of the COVID pandemic and their satisfaction with online teaching through the Zoom platform. The second part examines the satisfaction of students with Zoom teaching and learning process, i.e., its main advantages (better communication, not being conditioned by location related to attending classes, lower study costs and the possibility of recording classes) and disadvantages (impossibility of direct communication, technical problems, poor concentration and lack of digital competence) while the third part analyzes the effectiveness of communication methods in teaching and learning Mother Tongue Spelling (dialogue, monologue, textual and combined). The last part of the survey examines the contribution of Zoom to the improvement of knowledge, skills and competence related to *Mother Tongue Spelling.*

### 3.2. Data

Descriptive statistics were used for data analysis.

Of the total number of surveyed students, 77% come from the state university, 67% attend undergraduate studies, 19% are male, and only 8.3% did not regularly attend the course Mother Tongue Spelling via the Zoom application.

Considering that of the 600 students who study their mother tongue, the majority come from the state university and are at undergraduate level of studies, the sample is mostly proportional to the state of the population. Likewise, more women attend *Mother Tongue* studies than men (i.e., about 80% at the state university are women). Accordingly, the sample can be considered valid. Furthermore, the DT algorithm is completely insensitive to the specific values of the predictors [63], so their unbalanced distribution will not have negative effects on the data analysis.

By analyzing the results of the survey, it may be stated that out of the total number of respondents, 51.7% of respondents answered that they are generally satisfied with the Zoom learning of *Mother Tongue Spelling*, 38.7% partially, and about 9.6% said they are not satisfied. Approximately 75% of the respondents attending master studies stated that they were satisfied with the Zoom learning of *Mother Tongue Spelling*, while 25% of them were partially satisfied. Approximately 14% of respondents in undergraduate studies were not satisfied with Zoom learning *Mother Tongue Spelling*, about 40% were satisfied, while 46% were partially satisfied. The key reason why master's students were more satisfied with ZOOM classes than those from undergraduate studies was the fact that most master's students were employed, they found attending classes out of the classroom more suitable and they also had the possibility of listening to the recorded lessons later.

From private universities, 88% of students were satisfied with Zoom learning of *Mother Tongue Spelling*, compared to approximately 40% of the students from the state university. This result was expected because the majority of the students from the private university, due to their job engagement, had ZOOM classes even before the COVID pandemic, which was not the case with the students from the state university. The percentage of males and females who were satisfied with this kind of learning was equal.

Table 1 shows that there is a relation between the type of university (public/private) a student attends and the attitude about whether Zoom teaching improves literacy (*p*-value = 0.007). Furthermore, the attitude about the influence of Zoom on literacy depends on the level of study, full-time studying, teaching through Zoom experience and regular attendance at Zoom classes (*p*-values < 0.05). On the other hand, there is no statisti-

cally significant difference between the genders when it comes to the attitude towards the improvement of literacy using Zoom classes (*p*-value = 0.356).

**Table 1.** Students' profile and influence of the Zoom application on literacy.

| Students' Profile | Zoom Improves Literacy |
| --- | --- |
| Type of University | X-squared = 9.838, df = 2, *p*-value = 0.007 |
| Level of study | X-squared = 8.309, df = 2, *p*-value = 0.016 |
| Gender | X-squared = 2.065, df = 2, *p*-value = 0.356 |
| Full-time student | X-squared = 11.338, df = 2, *p*-value = 0.003 |
| Teaching through Zoom | X-squared = 25.666, df = 2, *p*-value = 0.000 |
| Regular attendance | X-squared = 24.283, df = 4, *p*-value = 0.000 |

Source: Authors' calculation.

The results of the survey also show that out of the total number of respondents, 50% of them stated that online teaching fully motivated them to self-initiatively collect materials from the reference field via the Internet, that is, 30% of them said that it partially motivated them. Approximately 90% of respondents believed that materials collected from the Internet, as well as classes conducted through the Zoom application, improved students' literacy, and 60% believe that it improved completely, and about 30% partially.

The survey shows that undergraduate students prefer the traditional type of learning, considering that they had the same type during their high school education and need a longer period of adaptation to the new type of learning, while blended learning is preferred in master studies, which is as expected because some of them are already employed.

Approximately 20% of male respondents in undergraduate studies, as well as 25% of females, prefer Zoom learning compared to traditional and blended, while in master studies, approximately 53% of males and 27% of females prefer Zoom learning compared to the other two models.

Based on the data from Figure 2, it is possible to obtain answers related to the advantages and disadvantages of Zoom learning of *Mother Tongue Spelling*.

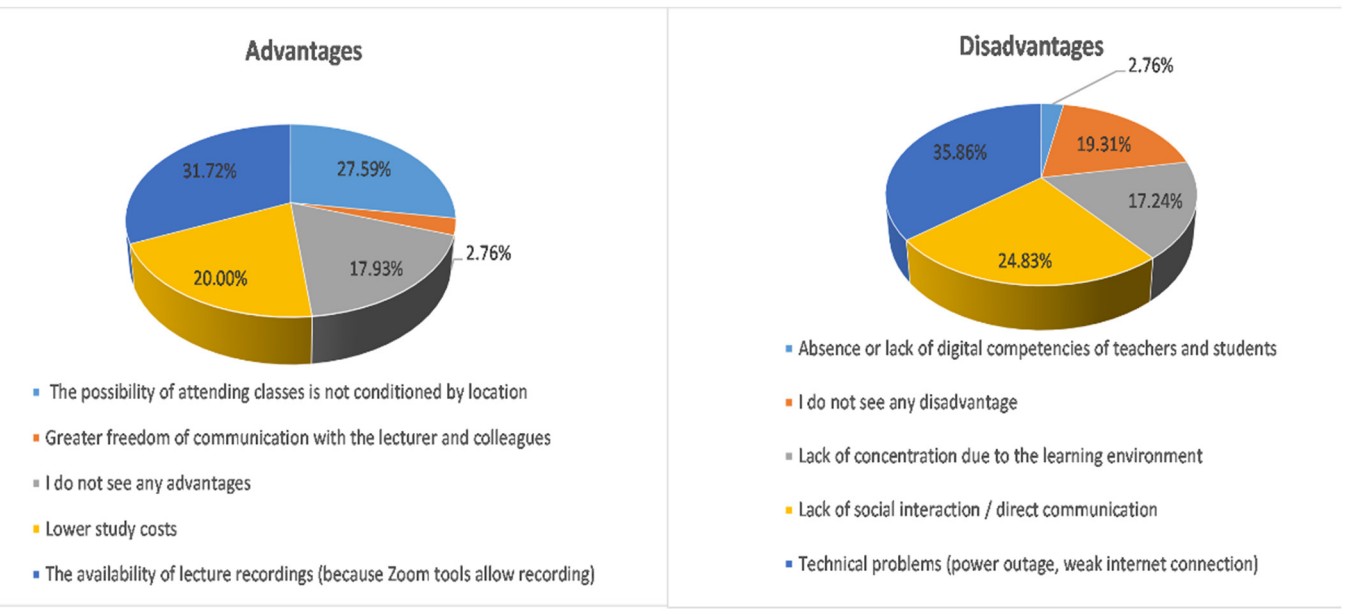

**Figure 2.** Disadvantages and advantages of Zoom learning of *Mother Tongue Spelling*.

Figure 2 shows that the largest number of respondents (31.7%) stated that the availability of lecture recordings is the biggest advantage of this type of teaching. After that, an important advantage is the possibility of attending classes from anywhere (location)

(27.6%) and, then, lower study costs when classes are conducted via Zoom. On the other hand, 35.7% of respondents recognized technical problems as the biggest limitation of Zoom learning *Mother Tongue Spelling*, as well as the lack of social interaction, i.e., direct communications (24.8%). It is important to emphasize that almost 20% of respondents believe that Zoom learning of Mother Tongue Spelling has no restrictions.

Based on the analysis of data from the survey, the answers of the respondents regarding the advantages and disadvantages of Zoom learning of Mother Tongue Spelling can be observed by clusters, i.e., by type of university, level of studies, and gender.

The largest number of respondents attending undergraduate studies at the state university stated that the biggest advantage of Zoom learning of Mother Tongue Spelling is the presence of teaching that is not conditioned by location, that is, they can attend classes from home, or any other place, and not only from the classroom. Such answers can be justified by the fact that many students do not live in the city where the classes are held, i.e., faculties are located in the central region of Montenegro and a large number of the students come from the northern and southern regions.

The largest number of respondents attending master studies at the state university stated that the biggest advantage is the availability of lecture recordings. This answer is also expected, given the fact that many students who attend master studies also work, and are not able to attend lectures in real-time.

When it comes to respondents of master studies from private universities, they identified lower study costs as the biggest advantage of Zoom learning of Mother Tongue Spelling, which is expected since students at private universities are exposed to higher study costs compared to state university students in Montenegro. Furthermore, it may be seen from the graph that the attitude regarding the advantages of Zoom learning Mother Tongue Spelling is similar among the male and female population at private universities, which is not the case with the state university. At the state university, there is a small number of men who attend classes in their mother tongue, and therefore, the conclusion regarding the gender cluster is not so relevant.

The largest number of respondents who attended undergraduate studies at the state university stated that technical problems were the dominant limitation of Zoom learning Mother Tongue Spelling, followed by a lack of direct communication. Additionally, none of the respondents from the undergraduate studies of the state university mentioned the lack of digital competencies as a limitation of this type of learning, while women from master's studies considered digital competence as a problem. When it came to respondents from private universities who attended classes related to Mother Tongue Spelling in their master studies, most of them stated that Zoom classes have no special restrictions. Such answers indicate significant IT literacy in Montenegrin students and the efficiency of the teaching and learning with Zoom during the pandemic. It may also be concluded that the lack of concentration is a key limitation of this type of teaching and learning process for the female population from private universities, as well as the lack of direct communication for the female population from the state university.

The following table may provide answers regarding the effects of Zoom teaching and learning of Mother Tongue Spelling regarding the teaching units, as well as the effectiveness of different methods in Zoom teaching and learning Mother Tongue Spelling.

Table 2 clearly shows that the implementation of the monologue method is the least effective in Zoom teaching and learning Mother Tongue Spelling.

On the other hand, according to the respondents' answers, the blended method proved to be the most effective.

The dialogical method proved to be the best for acquiring knowledge in the field of voice alternation in Zoom teaching and learning, which was pointed out by most of the students (38 of them).

The textual method proved to be the easiest in Zoom teaching and learning Mother Tongue Spelling for acquiring knowledge in the field of spelling and punctuation marks.

**Table 2.** Zoom teaching and learning Mother Tongue Spelling—the most effective methods for learning different units.

| | Methods | | | | |
|---|---|---|---|---|---|
| **Learning Units** | **Blended Method** | **Dialogical Method** | **Monologue Method** | **Text Method** | **Total** |
| The field of writing capital letters | 99 | 14 | 6 | 26 | 145 |
| % of total | 68.28% | 9.66% | 4.14% | 17.93% | 100.00% |
| The field of joint and separate word writing | 89 | 16 | 6 | 34 | 145 |
| % of total | 61.38% | 11.03% | 4.14% | 23.45% | 100.00% |
| The field of writing abbreviations | 90 | 16 | 11 | 28 | 145 |
| % of total | 62.73% | 11.03% | 6.93% | 19.31% | 100.00% |
| The field of voice alternation | 89 | 38 | 6 | 12 | 145 |
| % | 61.38% | 26.21% | 4.14% | 8.28% | 100.00% |
| The field of writing the consonants j and h: | 93 | 20 | 8 | 24 | 145 |
| % | 64.14% | 13.79% | 5.52% | 16.55% | 100.00% |
| The field of losing consonants | 91 | 26 | 10 | 18 | 145 |
| % | 62.76% | 17.93% | 6.90% | 12.41% | 100.00% |
| The field of spelling and punctuation | 89 | 12 | 8 | 36 | 145 |
| % | 61.39% | 8.28% | 5.52% | 24.83% | 100.00% |

Source: Authors' calculation.

## 4. Results

In order to answer the research questions, four DT models were generated, which indicate the relationships between the students' profile and their attitudes about the advantages (first model) and disadvantages of Zoom classes (second model), as well as about preferred methods of learning *Spelling* (third model) and teaching models (fourth model).

Although the exploratory data analysis determined that there is a difference between which category of students perceives the advantages of classes via Zoom, this result is statistically confirmed by the DT model in Figure 3.

This DT model provides an answer to RQ1, confirming that the perception of the advantages of Zoom lectures depends on the profile of the student. Reading the rules from the roots to the leaves of the tree (Figure 3), it can be concluded that those who are not full-time students do not see the advantages of this type of classes. Full-time students who are not satisfied with online classes via Zoom, which they attended occasionally, also do not see the advantages of this type of learning. Full-time students of the state university who are partially satisfied with Zoom classes, and attend them occasionally, see the "location" (attending classes from any place) as an advantage, while those who attend them regularly and are males, do not see particular advantages. These same students, including females from the master's studies of the state university, where teaching Mother Tongue Spelling was not predominantly organized online, see the availability of lecture recordings as an advantage of Zoom classes.

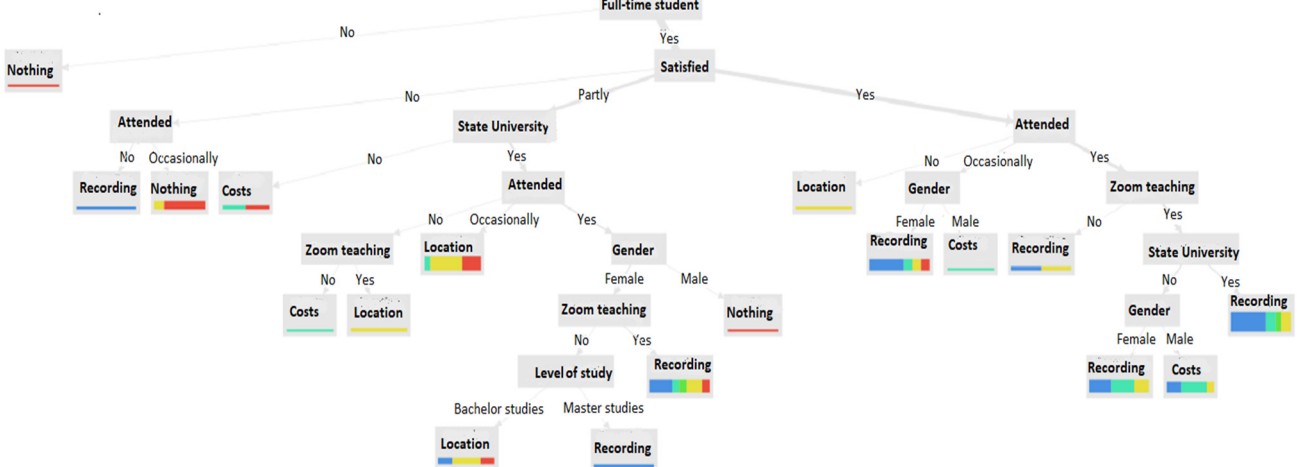

**Figure 3.** The relationship between the students' profiles and their perception of the advantages of Zoom lectures. Note: Each leaf (class defined by the dependent variable—advantages of ZOOM teaching) is assigned a corresponding colour (Nothing—red; Recording—blue; Costs—turquoise; Location—yellow). The most significant are the leaves in which the color to which the leaf belongs dominates.

Full-time students of the state university who are partially satisfied with Zoom classes, and who in fact did not attend the classes, believe that the "location" is an advantage of this type of teaching if teaching Mother Tongue Spelling is dominantly organized through Zoom, and if it is not, then they marked "costs" (lower study costs) as an advantage. Full-time students from the state university satisfied with Zoom classes, whose *Mother Tongue* classes were predominantly organized online and who attended them regularly, mostly see recorded lectures as an advantage. Full-time students, satisfied with Zoom classes, and who did not attend Mother Tongue Spelling classes, see "location" as an advantage, while those who partially attended them see "costs" as an advantage if they are males, as well as the possibility of recording classes if they are females.

The performance of the DT model from the previous picture is given through the confusion matrix in Table 3. It shows the precision of the model by individual classes as well as its overall accuracy.

**Table 3.** Classification performance of the DT model for the advantages of Zoom lectures.

|  | True Recording | True Costs | True Communication | True Location | True Nothing | Class Precision |
|---|---|---|---|---|---|---|
| pred. Recording | 40 | 14 | 4 | 16 | 4 | 51.28% |
| pred. Costs | 4 | 13 | 0 | 2 | 2 | 61.90% |
| pred. Communication | 0 | 0 | 0 | 0 | 0 | 0.00% |
| pred. Location | 2 | 2 | 0 | 20 | 8 | 62.50% |
| pred. Nothing | 0 | 0 | 0 | 2 | 12 | 85.71% |
| class recall | 86.96% | 44.83% | 0.00% | 50.00% | 46.15% | accuracy: 58.62% |

Table 3 shows that the overall accuracy of the model (the number of correctly classified data in relation to the total number of data) is about 60%, which means that the model correctly classified the majority of students. Additionally, by individual classes, the accuracy is higher than 50%, except for communication, which was cited as an advantage by an extremely small number of students, so the algorithm ignored this minor class.

To answer RQ2, the DT model in Figure 4 was generated.

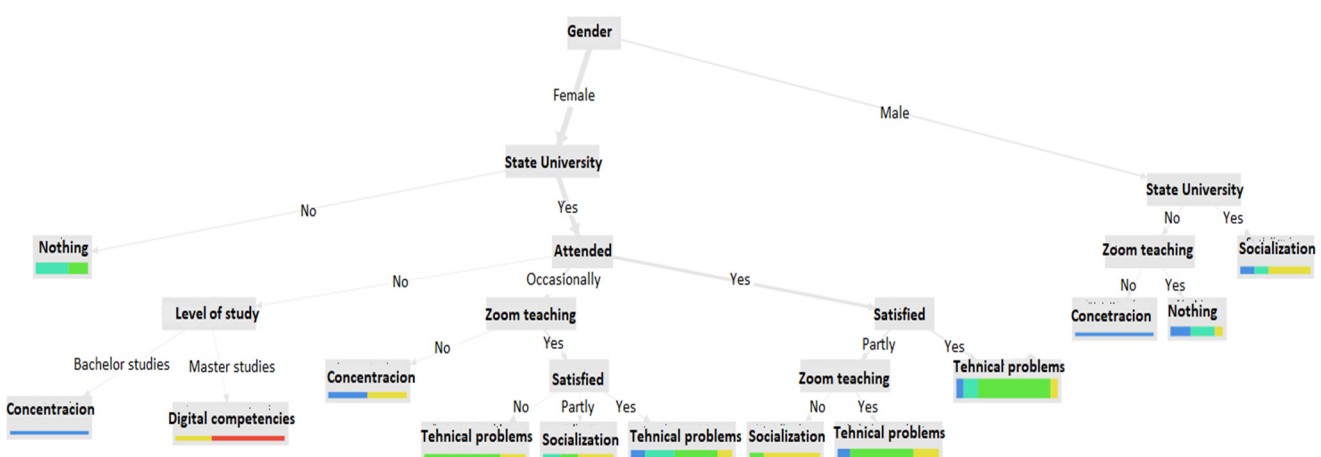

**Figure 4.** The relationship between the students' profiles and their perception of the disadvantages of Zoom classes. Note: Each leaf (class defined by the dependent variable—disadvantages of ZOOM teaching) is assigned a corresponding color (Concentration—blue, Nothing—turquoise; Digital competence—red, Technical problems—green, Socialization—yellow). The most significant are the leaves in which the color to which the leaf belongs dominates.

The DT model shown in Figure 4 gives us answers to RQ2 by analyzing the rules of the tree from tree to leaf. The DT model shown in Figure 4 gives us the answers to RQ2 by looking at the DT rules from the tree to the leaf. Female students who are not from the state university generally do not see the disadvantages of this type of teaching. If they are from the state university and do not attend Zoom classes and are at the bachelor's level, they state that the main disadvantage of Zoom classes is the difficulty of concentration, whereas master students point out their digital competence as the main problem. Female state university students who regularly attend Zoom classes, and who are completely satisfied with this type of teaching, mostly complain about technical problems, while those who are partially satisfied and whose *Mother Tongue* classes were not organized predominantly online see a lack of socialization as the main problem, i.e., lack of direct communication. Male state university students mostly state socialization as a lack or have no objections, while male private university students who did not have dominantly organized online *Mother Tongue* classes consider its main disadvantage to be poor concentration.

The validity of this DT model is confirmed by the confusion matrix in Table 4, where the algorithm correctly classified most of the examples in total and by lecture.

**Table 4.** Classification performance of the DT model for the disadvantages of Zoom teaching.

| | True Concentration | True Nothing | True Technical Problems | True Socializ | True Digital Competencies | Class Precision |
|---|---|---|---|---|---|---|
| pred. Concentration | 12 | 0 | 0 | 4 | 0 | 75.00% |
| pred. Nothing | 5 | 16 | 6 | 2 | 0 | 55.17% |
| pred. Technical problem | 6 | 8 | 42 | 10 | 0 | 63.64% |
| pred. Socialization | 2 | 4 | 4 | 18 | 0 | 64.29% |
| pred. Digital Competence | 0 | 0 | 0 | 2 | 4 | 66.67% |
| class recall | 48.00% | 57.14% | 80.77% | 50.00% | 100.00% | accuracy: 63.45% |

In order to answer RQ3 regarding the preferred method for learning spelling, data clustering using the k-means method was first performed.

This method views data as vectors in n-dimensional space and groups them based on their Euclidean distance, thus, forming homogeneous groups—clusters. The goal is for

the data within one cluster to be as similar as possible in terms of the characteristics by which the clustering is performed, and for the clusters to differ from each other as much as possible. Therefore, the number of clusters was chosen based on the Davies–Bouldin (DB) index [64], which compares the measure of homogeneity within clusters and heterogeneity between clusters. The smaller the DB index, the more favorable this relationship is.

The data are clustered here according to the preferred method for learning: writing capital letters, writing assembled and disassembled words, writing abbreviations, voice alternations, writing consonants j and h, losing consonants, spelling and punctuation marks. The optimal number of 4 clusters was determined based on the DB index in Table 5. It can be seen that the DB index is the smallest for k = 2, so the data were divided into 2 clusters at the first step. On the second step, the second smaller cluster was divided into 3 more because the DB index had the biggest drop when moving from 2 to 3 clusters.

**Table 5.** Determination of the optimal number of clusters based on the DB index (clustering by the spelling learning method).

|  | **K** | **2** | **3** | **4** | **5** |
|---|---|---|---|---|---|
| Step 1 | DB | 1.234 | 1.928 | 1.671 | 1.634 |
| Step 2 | DB | 2.365 | 1.926 | 1.943 | 1.550 |

Therefore, as a final result of clustering, 4 clusters were obtained (the first one was obtained by initial clustering—step 1, and the other three by re-clustering of the second cluster—step 2) whose centroids are shown in Table 6.

**Table 6.** Centroids of clustering according to the preferred method of learning *Spelling*.

| Attribute | Cl0 (87) [1] | Cl1 (24) | Cl2 (26) | Cl3 (8) |
|---|---|---|---|---|
| Monologue method = writing capital letters | 0.000 | 0.083 | 0.000 | 0.500 |
| Monologue method = joint and separate word writing | 0.000 | 0.083 | 0.077 | 0.250 |
| Monologue method = voice alternation | 0.000 | 0.167 | 0.000 | 0.250 |
| Monologue method = writing the consonants j and h | 0.000 | 0.000 | 0.000 | 0.750 |
| Monologue method = losing consonants | 0.023 | 0.083 | 0.000 | 0.750 |
| Monologue method = spelling and punctuation | 0.000 | 0.083 | 0.000 | 0.750 |
| Monologue method = writing abbreviations | 0.000 | 0.167 | 0.000 | 0.750 |
| Dialogical method = writing capital letters | 0.069 | 0.000 | 0.308 | 0.000 |
| Dialogical method = joint and separate word writing | 0.069 | 0.167 | 0.154 | 0.250 |
| Dialogical method = voice alternation | 0.115 | 0.250 | 0.769 | 0.250 |
| Dialogical method = writing the consonants j and h | 0.023 | 0.000 | 0.615 | 0.250 |
| Dialogical method = losing consonants | 0.069 | 0.083 | 0.692 | 0.000 |
| Dialogical method = spelling and punctuation | 0.000 | 0.000 | 0.385 | 0.250 |
| Dialogical method = writing abbreviations | 0.000 | 0.083 | 0.538 | 0.000 |
| Blended method = writing capital letters | 0.839 | 0.417 | 0.538 | 0.250 |
| Blended method = joint and separate word writing | 0.908 | 0.167 | 0.231 | 0.000 |
| Blended method = voice alternation | 0.862 | 0.167 | 0.231 | 0.500 |
| Blended method = writing the consonants j and h: | 0.023 | 0.083 | 0.308 | 0.000 |
| Blended method = losing consonants | 0.908 | 0.167 | 0.231 | 0.250 |
| Blended method = spelling and punctuation | 0.977 | 0.000 | 0.154 | 0.000 |

**Table 6.** *Cont.*

| Attribute | Cl0 (87) [1] | Cl1 (24) | Cl2 (26) | Cl3 (8) |
|---|---|---|---|---|
| Blended method = writing abbreviations | 0.977 | 0.083 | 0.077 | 0.250 |
| Text method = writing capital letters | 0.092 | 0.500 | 0.154 | 0.250 |
| Text method = joint and separate word writing | 0.023 | 0.583 | 0.538 | 0.500 |
| Text method = voice alternation | 0.023 | 0.417 | 0.000 | 0.000 |
| Text method = writing the consonants j and h | 0.954 | 0.917 | 0.077 | 0.000 |
| Text method = losing consonants | 0.000 | 0.667 | 0.077 | 0.000 |
| Text method = spelling and punctuation | 0.023 | 0.917 | 0.462 | 0.000 |
| Text method = writing abbreviations | 0.023 | 0.667 | 0.385 | 0.000 |
| Method | **Blended** | **Text** | **Dialogical** | **Monologue** |

[1] The number in brackets represents the number of students in the cluster.

Based on the bold centroids, it can be concluded that in cluster 0, the majority of students prefer the blended learning method for all teaching units. In cluster 1, the largest centroids are for the text method for all units, so there are students who prefer this method. Cluster 2 consists mainly of students who prefer the dialogic method for most units, and cluster 3 is monologic. In this way, each student is classified into one of these 4 clusters, which defines the method of learning Mother Tongue Spelling that he/she prefers.

Now it is possible to classify students according to these 4 clusters, i.e., methods. For this purpose, the DT model in Figure 5 was generated.

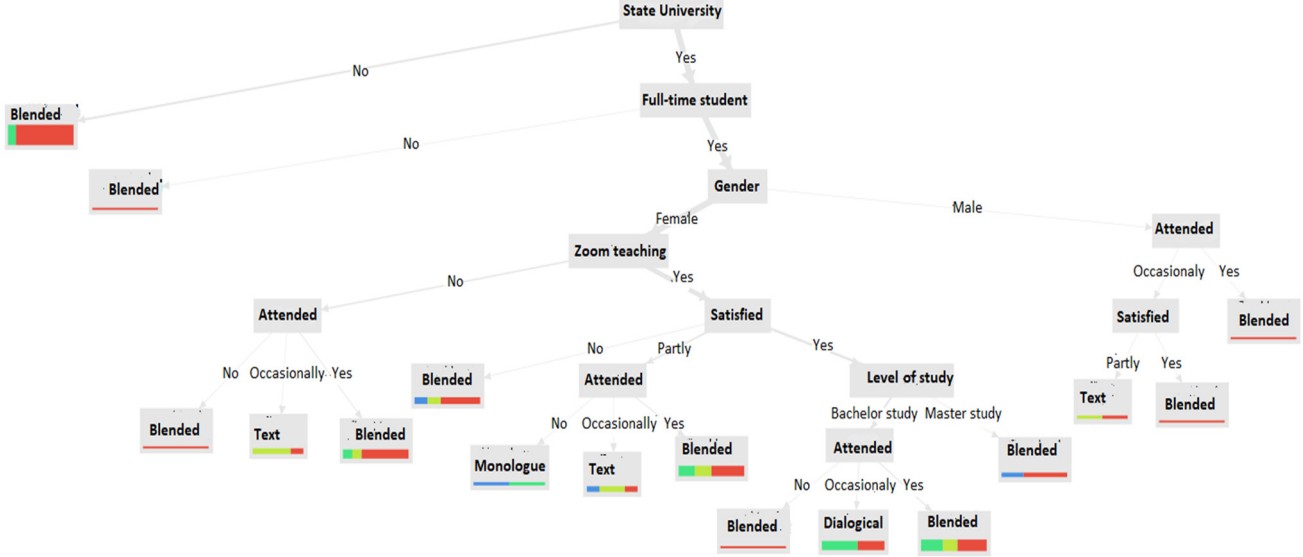

**Figure 5.** The relationship between the students' profile and the methods for teaching Mother Tongue Spelling they prefer during Zoom lessons. Note: Each leaf (class defined by the dependent variable–methods) is assigned a corresponding color (Blended—red; Text—yellow; Monological—blue; Dialogical—green).The most significant are the leaves in which the color to which the leaf belongs dominates.

The answer to RQ3 can be found in the analysis of the DT tree shown in Figure 5, which clearly indicates that the majority of respondents prefer the blended method when it comes to online teaching *Mother Tongue Spelling*; private university students and part-time state university generally prefer the blended method for learning *Spelling*. Full-time female state university students whose Mother Tongue Spelling classes were predominantly organized online via Zoom, and who were partially satisfied with the classes, prefer the

textual method for learning Spelling if they partially attended them, as well as the blended method if they regularly attended classes. Full-time female state university students from master's studies whose Mother Tongue Spelling classes were predominantly organized online via Zoom and who were completely satisfied with this type of teaching prefer the blended method, as well as the largest number of the undergraduate students who did not attend the classes regularly. Full-time female state university students whose Mother Tongue Spelling classes were not predominantly organized online via Zoom, and who generally did not attend them via the Zoom platform, also preferred the blended method. Male full-time state university students, who regularly or occasionally attended online classes, mostly preferred the blended method, while only those who were partially satisfied with this type of teaching preferred the textual method.

The validity of this DT model is confirmed by the confusion matrix in Table 7.

**Table 7.** Classification performance of the DT model for *Mother Tongue Spelling*.

|  | True Monologue | True Dialogical | True Text | True Combined (Blended) | Class Precision |
|---|---|---|---|---|---|
| pred. Monologue | 2 | 2 | 0 | 0 | 50.00% |
| pred. Dialogical | 0 | 8 | 0 | 6 | 57.14% |
| pred. Text | 2 | 0 | 12 | 6 | 60.00% |
| pred. Combined (blended) | 4 | 16 | 12 | 75 | 70.09% |
| class recall | 25.00% | 30.77% | 50.00% | 86.21% | accuracy: 66.90% |

Finally, to answer RQ4 regarding the preferred way of learning, students were first clustered based on the following characteristics: whether online teaching motivated them to search for materials on the Internet, whether they thought that materials collected from the Internet improved their spelling knowledge, whether learning with Zoom helped them improve their spelling skills, and which teaching method they preferred (Zoom, traditional or blended). The optimal number of cluster 3 was determined based on the DB index (Table 8).

**Table 8.** Determination of the optimal number of clusters based on the DB index (clustering by teaching method).

| K | 2 | 3 | 4 | 5 |
|---|---|---|---|---|
| DB | 1.774 | 1.349 | 1.213 | 1.246 |

The following Table 9 shows the centroids for three clusters, i.e., the points whose coordinates are the average values of the coordinates of all points belonging to the cluster.

From the table, based on the bold centroids, the students from cluster 0 believed that online teaching motivated them to find materials on the Internet, that these materials improved their spelling knowledge, and that learning via Zoom improved spelling and orthographic habits; therefore, this cluster is said to prefer instructions via Zoom. The students from cluster 1 believed that all this was partially true and preferred blended classes, while the students from cluster 2 denied the importance of online classes for motivation and improving spelling knowledge and skills, so they preferred traditional instruction.

Now it is possible to classify these 3 groups of students, so for this purpose, the DT model in Figure 6 was generated.

**Table 9.** Centroids of clustering by preferred method of teaching.

| Attribute | Cl0 (73) [1] | Cl1 (38) | Cl2(34) |
|---|---|---|---|
| materials on the Internet = Yes | 0.781 | 0.368 | 0.000 |
| materials on the Internet = Partly | 0.219 | 0.632 | 0.059 |
| materials on the Internet = No | 0.000 | 0.000 | 0.941 |
| internet materials improved spelling knowledge= Yes | 1.000 | 0.000 | 0.294 |
| internet materials improved spelling knowledge= Partly | 0.000 | 1.000 | 0.059 |
| internet materials improved spelling knowledge = No | 0.000 | 0.000 | 0.647 |
| Zoom improved spelling and orthography habits= Partly | 0.055 | 0.632 | 0.412 |
| Zoom improved spelling and orthography habits = Yes | 0.863 | 0.263 | 0.235 |
| Zoom improved spelling and orthography habits= No | 0.082 | 0.105 | 0.353 |
| type of instruction = Blended | 0.356 | 0.474 | 0.118 |
| type of instruction = Traditional | 0.192 | 0.368 | 0.824 |
| type of instruction = Via Zoom | 0.452 | 0.158 | 0.059 |
| Instructions | **Via Zoom** | **Blended** | **Traditional** |

[1] The number in brackets represents the number of students in the cluster.

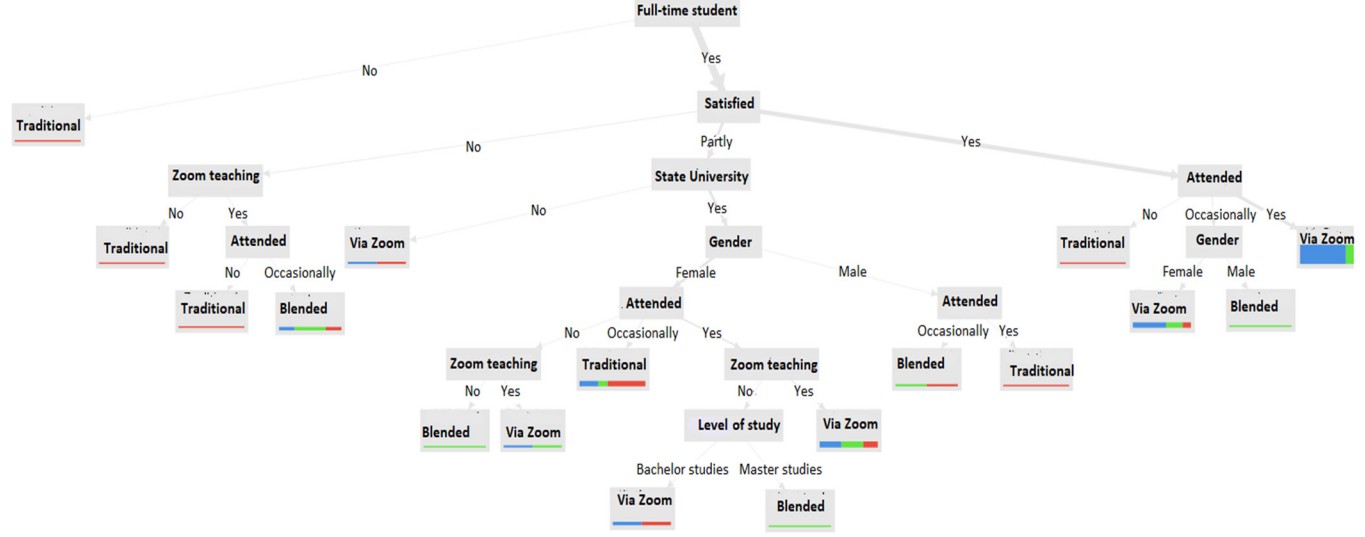

**Figure 6.** Connection between the students' profiles and the preferred type of teaching. Note: Each leaf (class defined by the dependent variable–type of teaching) is assigned a corresponding color (Traditional—red; Blended—green; Via Zoom—blue). The most significant are the leaves in which the color to which the leaf belongs dominates.

By reading the rules from the roots to the leaves of the tree (Figure 6) we obtain the answer to RQ4. Part-time students preferred traditional classes, as well as a significant number of full-time students who were not satisfied with Zoom classes, whose Mother Tongue Spelling classes were predominantly organized through the Zoom platform, and who did not attend them regularly. The same was the case with full-time students who were satisfied with Zoom classes, but who did not attend online Mother Tongue Spelling classes, while students who attended Zoom classes related to this course saw this as an advantage. Full-time students who were satisfied with online classes but attended them occasionally gave preference to Zoom classes if they were females or blended teaching if they were males. Full-time female state university students who were partially satisfied with online classes and attended them occasionally preferred traditional classes, while those who regularly attended them even though classes were not predominantly organized

via Zoom preferred classes via Zoom (bachelor level) or blended (master level). Full-time female state university students who were partially satisfied with online classes, and whose Mother Tongue Spelling classes were not predominantly organized online, and who did not attend them regularly, preferred the blended method. Their male colleagues who were also partially satisfied with online teaching and attended Zoom classes regularly or occasionally preferred traditional or blended classes. Finally, full-time students who were partially satisfied with online classes and came from a private university, mostly preferred classes via Zoom.

The classification performance that confirms the validity of this DT model is given in Table 10.

**Table 10.** Classification performance of DT model for the preferred method of teaching.

|  | True via Zoom | True Combined (Blended) | True Traditional | Class Precision |
|---|---|---|---|---|
| pred. Via Zoom | 67 | 24 | 12 | 65.05% |
| pred. Blended | 2 | 12 | 4 | 66.67% |
| pred. Traditional | 4 | 2 | 18 | 75.00% |
| class recall | 91.78% | 31.58% | 52.94% | accuracy: 66.90% |

## 5. Discussion

The DT models as shown in figures in the previous chapter will provide a detailed analysis of the respondents' answers depending on their profile. This chapter will explain each RQ and the most important results of the analysis will be presented in summary, as key answers to each set RQ.

Looking at the most significant advantages of online teaching, that is, in order to answer RQ1, the results show that the perception of the advantages of Zoom lectures depends on the profile of the student. It can be concluded that full-time students who do not perceive the advantage of Zoom classes are predominantly those who attend it only occasionally and are not satisfied with it, as well as male state university students who are partially satisfied with these classes and attend them regularly. State university students who are partially satisfied with the Zoom spelling classes and attend them occasionally, or do not attend them regularly, even though Mother Tongue Spelling classes are predominantly organized through the Zoom platform, prefer "location". The possibility to record classes, as an advantage of Zoom classes, is seen by full-time female state university students from master's studies who are partially satisfied with the Mother Tongue Spelling classes they attended, although they were not dominantly organized online, as well as full-time state university students who are satisfied with Zoom spelling classes, which were dominantly organized online and who attended them regularly. The same perception is shared by full-time female students who occasionally attend Zoom classes and are satisfied with them. Full-time state university students who are partially satisfied with Zoom classes, but who do not attend them, as well as students whose Mother Tongue Spelling classes were not predominantly organized online, see "costs" as an advantage. Full-time students who are satisfied with Zoom classes, and who occasionally attended them share the same attitude. Such findings can be explained by the fact that in Montenegro, for young women who are studying and simultaneously taking care of the family, i.e., raising children, this type of learning provides the opportunity to obtain an education but also to spend most of their time with their children, which would not be possible in the case of live classes, since they do not live in the city where they study in most cases. That is why they prefer Zoom classes, because they can attend them from home or listen to recorded lectures in their spare time. Furthermore, a large number of students from the northeastern and southern regions rent an apartment during their studies or are forced to pay travel fees because they are studying outside their place of residence, e.g., in the central region where the universities are located, so they perceive saving money as a key advantage.

Similar results were obtained by a group of authors who conducted their research at Romanian faculties, where the authors showed that "location" is the main advantage of online classes, because students can attend classes from home or any other place, which implies saving time and money [43]. These authors showed that those advantages could help design courses that fit the needs of certain categories of students (those who work, who are unable to attend courses, who cannot afford to study in another city, etc.). Additionally, that "location" is the main advantage of this type of teaching is stated in the paper [47]. The authors of a study conducted with Polish medical students reached similar results regarding the advantages of online teaching [44]. That "costs" are one of the advantages of online classes was also confirmed by the authors who examined the students' perception about online classes during COVID-19 [32]. Other authors confirmed that the main advantage of online teaching is that it offers students the opportunity to learn when they want, as Zoom provides a recording tool [46].

Based on the analysis of students' answers related to their dissatisfaction with online teaching, we obtain the answers to RQ2, which, among other things, show that female private university students do not perceive the disadvantages of this type of teaching, and those females who are undergraduates at the state university and did not attend Zoom classes see a lack of concentration as the main disadvantage. Male private university students whose Mother Tongue Spelling classes were not dominantly organized online also consider poor concentration to be the main disadvantage of this type of teaching. Digital competence is the biggest shortcoming of this teaching format, according to female state university master studies students who did not attend classes regularly. Female state university students who regularly attended Zoom classes and who are completely satisfied with them primarily point out technical problems as the main disadvantage, while those who are partially satisfied and whose Mother Tongue Spelling classes were not conducted online too often consider the lack of direct communication between students as the main problem, i.e., inadequate socialization. Male state university students point out the same problem. The lack of concentration and insufficient socialization mentioned by the majority of respondents, both males and females, are an indicator of the slow adaptation to the new type of teaching compared to the traditional one, which dominated before the COVID-19 pandemic in our country. Students single out technical problems as well, which is not surprising considering that Montenegro is a developing country and students are not adequately equipped technically.

Both Romanian and Polish students surveyed [43,44] stated that poor interaction (lack of direct communication), as well as technical problems, are the main disadvantages of online teaching. The same conclusion was reached by the authors who examined the perception of students in Pakistan about online learning during the COVID-19 pandemic [45]. Furthermore, students from central European universities state that poor social interaction is one of the biggest disadvantages of online teaching in foreign language learning [47].

As for RQ3, we can conclude that private university students, as well as part-time state university students, prefer the blended teaching method. Female full-time state university students whose Mother Tongue Spelling classes were predominantly organized via Zoom, and who regularly attended classes and were partially satisfied with them, also preferred the blended method. The same method was also preferred by full-time state university students in master's studies, who were completely satisfied with this type of teaching. The blended method was also preferred by female students from undergraduate studies, for whom Mother Tongue Spelling classes were organized online. Most students, regardless of the profile they belong to, preferred the blended method. Only part of the students who were partially satisfied with online teaching through the Zoom application, and who came from the state university, preferred the text method. Such results can be justified by the fact that the blended method breaks the monotony of monologues, all participants in the teaching process are actively engaged, both the lecturer and the students, thus, intensifying oral and written communication necessary for successful mastering of literacy

skills. Therefore, Figure 5 clearly indicates that the majority of respondents prefer the blended method when it comes to online teaching *Mother Tongue Spelling*.

Studies based on the teaching experience of lecturers in the online and traditional model of mother tongue teaching confirm that combining communication methods is the most effective way to achieve teaching objectives [24], and that "valid teaching practice is based on innovative approach" [23]. We must mention once again that the research on the effects of the application of communication methods during Zoom classes was not a subject of scientific interest, and that analogies can only be established with the general comments individually given by the authors who conducted research on online classes. In this regard, one of the authors emphasized the importance of the lecturer's communication skills, as well as his/her ability to use multimedia content with the aim of a successful presentation [46], which, when it comes to spelling courses, actually represents the lecturer's ability to combine adequate communication methods in a timely manner (dialogue, monologue and text) in delivering a certain teaching unit.

The answers to RQ4 show that traditional classes are preferred by part-time students, as well as a large number of full-time students whose Mother Tongue Spelling classes were predominantly organized via Zoom and who were not satisfied with this type of class and did not attend them regularly. The same case was with full-time students who were satisfied with Zoom lectures but who did not attend online Mother Tongue Spelling classes. Male full-time students who were satisfied with online classes, but attended them occasionally, also opted for the blended type of class, as well as female state university students at master's studies, who were partially satisfied with online classes, which they attended regularly but whose Mother Tongue Spelling classes were not dominantly organized via Zoom. Zoom classes were preferred by full-time students who are satisfied with them and who attended spelling lectures in this online format, as well as female full-time students who occasionally attended and were satisfied with this type of teaching. In general, Zoom teaching was preferred by students who were partially or completely satisfied with it, while students who were not satisfied with this type of teaching, as well as part-time students, preferred the traditional type of teaching. This is due to the fact that part-time students are predominantly of the older generation and are actually not familiar with the advantages of this type of teaching because they have no experience with it.

The subject of an empirical study was the attitudes of performing arts students in Hong Kong concerning teaching models: blended, online and face-to-face. The authors' findings are similar to ours when it comes to theory courses: undergraduate and graduate students prefer the blended and online models. When it comes to practical teaching, i.e., tutorials of stage dance performance, with which we cannot establish an analogy due to the curriculum and its diversity, students give preference to the traditional model. The authors also note that the surveyed students find the online model more suitable for language courses than for performing arts courses [65].The results of a study analyzing the perception of technology students show that they prefer the online model as it enables them to learn anywhere and anytime [66]. According to a survey of Pakistani students, only 10.3% of respondents claimed that online learning is motivating, and in this sense, they preferred the traditional teaching model as more effective, which the authors explain by the fact that in underdeveloped countries, such as Pakistan, most students do not have access to the Internet due to technical and financial reasons [45].

## 6. Conclusions

Our research showed that as many as 90% of the total number of surveyed students are satisfied or partially satisfied with online teaching of Mother Tongue Spelling conducted through the Zoom application. Based on the results, it can be concluded that the perception of the advantages and disadvantages of this teaching format depends on the students' profiles.

The greatest number of respondents (approximately 60%) stated that the possibility to record lectures and attend them from any place ("location") were the biggest advantages

of Zoom Mother Tongue Spelling classes, while they perceived technical problems and lack of social interaction as key disadvantages. The majority of respondents considered the blended method the most effective, while the monologist method was the least desirable for them. We also concluded that, according to students' perception, Zoom teaching motivated or partially motivated (80%) students to search for materials related to spelling via the Internet, i.e., to work independently, which ensures the continuity and sustainability of this type of teaching. As many as 90% of respondents claimed that teaching spelling through the Zoom platform improved literacy.

According to the research results, the answers to RQ1 as well as RQ2 are affirmative, that is, the perception of the advantages and disadvantages of this teaching format depends on the students' profiles. As for RQ3, we can conclude that private university students as well as part-time state university students prefer the blended teaching method. The answers to RQ4 show that the preferred model of teaching depends on the students' profiles.

As for the sustainability of online teaching *Mother Tongue Spelling*, having analyzed the research results, the authors conclude that embedding sustainability in higher education learning can be easily achieved and executed by utilizing the Zoom platform as an effective e-learning method, which has proven to have potential to be maintained, used and further developed.

Having in mind the importance of the implementation of online tools in the higher education teaching process, especially in the COVID-19 pandemic and the fact that this is the first research of this kind not only in Montenegro but generally in the Western Balkan countries, the results of the survey may be useful to many stakeholders in higher education—policy makers, relevant bodies and institutions in higher education, management of higher education institutions, teaching staff, students, etc. In addition to its practicality, the authors believe that this paper may provide a significant theoretical contribution. There is an obvious lack of literature in the English language regarding Montenegrin—Serbian, Croatian, Bosnian as a mother tongue, and this paper may fill this literature gap. Additionally, it may encourage other scientists from the other Western Balkan countries to publish papers and conduct research that deal with Montenegrin—Serbian, Croatian, Bosnian as a mother tongue in the English language.

Still, this research has some obvious limitations, especially regarding the socio-demographic component of the sample, as well as the fact that the multiple choice questions were not included in the survey. It was also possible to include the students' comments and explanations for the selection of certain answers in the survey, i.e., the study would be more comprehensive if it included the attitude of the students, as well as the professors, on the objectivity of the answers and evaluation within the online teaching model. For this reason, we strongly believe that, in the future, more detailed research should be conducted.

The philology students from other Western Balkan countries could be included in the research in order to compare the results in different countries and obtain more relevant conclusions regarding this topic In addition, this paper will benefit further research into the perception of communication methods in online teaching not only linguistic but also social-science-and-humanities courses, as well as the conditions and possibilities of its sustainability.

**Author Contributions:** Conceptualization, M.B. and M.N.B.; methodology M.B., M.N.B. and L.K.; software, L.K. and M.N.B.; validation, M.B. and A.J.S.; formal analysis, M.B, A.J.S. and D.Z.; investigation, M.B., M.N.B. and A.J.S.; resources, A.J.S. and M.B.; data curation, M.B. and A.J.S.; writing—M.B., M.N.B. and L.K.; writing—review and editing, A.J.S., M.B., L.K. and D.Z.; visualization, M.N.B. and D.Z. All authors have read and agreed to the published version of the manuscript.

**Funding:** This research received no external funding.

**Institutional Review Board Statement:** Not applicable.

**Informed Consent Statement:** Not applicable.

**Data Availability Statement:** The data presented in this study are available on request from the corresponding author. The data are not publicly available because they were used exclusively for the purposes of this research.

**Conflicts of Interest:** The authors declare no conflict of interest.

**Appendix A. Survey**

Teaching staff from the University of Montenegro together with their colleagues from University Donja Gorica carried out the scientific research regarding the implementation of communication methods in Zoom teaching and learning Mother Tongue Spelling at Montenegrin Universities.

All collected survey data will be considered confidential. They will serve to provide the final results. The contents of the survey will be treated discreetly and exclusively for the purpose described. We kindly ask you for your full cooperation. It is desirable that the survey be completed by students of the Faculty of Philology—department of Serbian/Montenegrin language.

1. Are you a student of the state university?
   (a) yes
   (b) no

2. You are a student of:
   (a) Bachelor studies
   (b) master studies

3. Gender:
   (a) male
   (b) female

4. Are you a full-time student?
   (a) yes
   (b) no

5. Have you regularly attended the Mother Tongue Spelling classes with Zoom app?
   (a) yes
   (b) no
   (c) occasionally

6. Have the Mother Tongue Spelling classes been predominantly conducted through the Zoom app in the previous two years?
   (a) yes
   (b) no

7. Are you satisfied with the teaching and learning Mother Tongue Spelling through the Zoom app?
   (a) yes
   (b) no

8. What is the most significant advantage of Zoom teaching and learning *Mother Tongue Spelling*:
   (a) the availability of lecture recordings (because Zoom tools allow recording)
   (b) the possibility of attending classes is not conditioned by location
   (c) greater freedom of communication with the lecturer and colleagues
   (d) lower study costs
   (e) I do not see any advantages

9. What is the most important disadvantage of Zoom teaching and learning *Mother Tongue Spelling*:
   (a) lack of social interaction/direct communication
   (b) technical problems (power outage, weak internet connection)

(c)　absence or lack of digital competencies of teachers and students
(d)　lack of concentration due to the learning environment
(e)　I do not see any disadvantage

10.　In attending the classes with Zoom, which method do you find the most appropriate to acquire knowledge, skills and competences in writing capital letters

(a)　dialogical method
(b)　monologue method
(c)　text method
(d)　blended method

11.　In attending the classes with Zoom, which method do you find the most appropriate to acquire knowledge, skills and competences in joint and separate word writing:

(a)　dialogical method
(b)　monologue method
(c)　text method
(d)　blended method

12.　In attending the classes with Zoom, which method do you find the most appropriate to acquire knowledge in writing abbreviations:

(a)　dialogical method
(b)　monologue method
(c)　text method
(d)　blended method

13.　In attending the classes with Zoom, which method do you find the most appropriate to acquire knowledge, skills and competences in voice alternation:

(a)　dialogical method
(b)　monologue method
(c)　text method
(d)　blended method

14.　In attending the classes with Zoom, which method do you find the most appropriate to acquire knowledge, skills and competences in writing the consonants j and h:

(a)　dialogical method
(b)　monologue method
(c)　text method
(d)　blended method

15.　In attending the classes with Zoom, which method do you find the most appropriate to acquire knowledge, skills and competences in losing consonants:

(a)　dialogical method
(b)　monologue method
(c)　text method
(d)　blended method

16.　Attending the classes through Zoom, which method is the most appropriate to acquire knowledge, skills and competences in spelling and punctuation:

(a)　dialogical method
(b)　monologue method
(c)　text method
(d)　blended method

17.　Have the online classes motivated you to search for materials related to Mother Tongue Spelling on the Internet on your own initiative?

(a)　Yes
(b)　No
(c)　partially

18. Have the materials collected from the Internet improved your spelling knowledge?

    (a) yes
    (b) no
    (c) partially
    (d) I did not search and collect materials on the Internet on my own initiative

19. Has learning through the Zoom app helped you become more literate, i.e., improve your spelling and orthography habits?

    (a) yes
    (b) no
    (c) partially

20. Which type of the learning Mother Tongue Spelling do you prefer:

    (a) traditional
    (b) via Zoom
    (c) blended

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
