# Peer review of "Sustainability of Online Teaching: The Case Study Mother Tongue Spelling Course at Montenegrin Universities"

_sustainability, doi:10.3390/su142113717_

Round 1

Reviewer 1 Report

I appreciate that the study addressed an important topic but I am not entirely sure of its knowledge contributions to the field. I think that the authors should address the following issues:

1. Literature review should be presented as a separate section from the introduction and methodology. The literature review has engaged with the issues such as sustainability and provided the contextual background. It does not engage with the relevant literature on the use of technology in language education. For this reason, it is not clear to this reader why the research questions presented in the methodology section had to be addressed in the inquiry. Why did the study need to look the connections between language learners' profiles and their perceptions/views? What do the authors mean by profiles? Why did the authors include certain characteristics into the participants' profiles, not others? 

2. I am not entirely sure if the DT model makes the findings more accessible for readers. I think that you need to articulate what answers they came up with in response to each research question in the analysis.

3. How do the findings constitute knowledge contributions to the field? I think that the authors need to engage with relevant studies on the use of technology in language education (especially those to do with online language learning and teaching) to argue how the findings of this study add to what we have already known about the topic. Please check out https://www.sciencedirect.com/journal/system/special-issue/10HCDZ0HBNR. 

Reviewer 2 Report

Dear authors,

I appreciate your work. However, a proofreading of the article indicates that you can improve it, and here you can find some suggestions about it.  

- Please, provide a short review of Zoom platform (generic functioning, advantages and disadvantages of its use) into the first part of the article.

Lines 55-58 – Please, provide some scientific reference to support your sentences, especially: “it enables students to improve their written and oral communication skills”. The point can be controversial.

Line 69 – It should be "Because"

Line 111-112 - Check the sentences. 

Line 184 - Check the spelling of the article: close square brackets.

Lines 225-247 - Please, anticipate in the introduction this explanation about Montenegrin spelling. So the text will be clearer for readers. 

Moreover, please provide general proofreading of the entire article in order to eliminate spelling errors and omissions. It is even more important considering that you talk about spelling yourself. 

Reviewer 3 Report

The first sentence in in the introduction is too long and creates confusion. Split the phrase in two to make it clearer. Do not use abbreviations in the abstract and keep the abstract simple, referring to the main objective of the paper, the method used and the findings. You can include a mention to the usefulness or the novelty of your study too. The rest are for the introduction or for the Methodology section. Keep the abstract simple and clean. 

 Literature review should be Heading no 2 not a subheading in the introduction

Line 146: References 15,16,17 should be [15-17]. Check for similar situations and correct. Line 331 the same. Line 343: Do not put an ending point before the reference but after. So, (one class). [53] should be (one class) [53].

Line 300: Use research methodology instead of methodology of research

All figures should be redrawn because the text is blurry and they seem to be of low quality

Line 683. Elaborate on your idea. Is too rough. 

Discussion should be separate than results. In the Discussion section you should reiterate your findings but in comparison with other papers and studies in the literature. 

The first part in the conclusions is in fact a Discussion when you present your RQs. But add references that support or not your findings. 

In the conclusion you should keep the theoretical and practical implications of your paper (usefulness), the limitations of your research and the future research directions especially if you consider the specific nature of your study. 

Round 2

Reviewer 1 Report

I think that you have addressed my concerns through revision. I have no further issue with your work.